# Pretrained Language Model in Continual Learning: A Comparative Study

**Tongtong Wu**[1,2]**, Massimo Caccia**[3]**, Zhuang Li**[2]**, Yuan-Fang Li**[2]**, Guilin Qi**[1]**, Gholamreza Haffari** [2]

[1]Southeast University [2]Monash University [3]MILA

`{wutong8023, gqi}@seu.edu.cn, massimo.p.caccia@gmail.com`
`{tongtong.wu, zhuang.li, yuanfang.li, gholamreza.haffari}@monash.edu,`

## Abstract

Continual learning (CL) is a setting in which a model learns from a stream of incoming data while avoiding to forget previously learned knowledge. Pre-trained language models (PLMs) have been successfully employed in continual learning of different natural language problems. With the rapid development of many continual learning methods and PLMs, understanding and disentangling their interactions become essential for continued improvement of continual learning performance. In this paper, we thoroughly compare the continual learning performance over the combination of 5 PLMs and 4 CL approaches on 3 benchmarks in 2 typical incremental settings. Our extensive experimental analyses reveal interesting performance differences across PLMs and across CL methods. Furthermore, our representativeness probing analyses dissect PLMs' performance characteristics in a layer-wise and task-wise manner, uncovering the extent to which their inner layers suffer from forgetting, and the effect of different CL approaches on each layer. Finally, our observations and analyses open up a number of important research questions that will inform and guide the design of effective continual learning techniques.

## 1 Introduction

Continual Learning (CL) methods aim at training a model from a stream of non-i.i.d. samples, relieving catastrophic forgetting (CF) while limiting computational costs and memory footprint. Throughout the years, many methods have been proposed to address the CL problem in computer vision and robotics (Kirkpatrick et al., 2017; Serrà et al., 2018; Buzzega et al., 2020). CL in NLP is still a nascent topic, as reflected by the relatively smaller number of proposed methods (Biesialska et al., 2020; Li et al., 2021). However, it is not always easy to precisely measure the merits of these works. This is partly due to the subtle differences in the way methods are evaluated: many state-of-the-art approaches only stand out in the setting where they were originally conceived. Moreover, pretrained language models (PLMs) have been widely applied in CL methods in NLP, and their addition further complicates the clear understanding of model performance (Han et al., 2020; Wang et al., 2019; Cao et al., 2020).

Although many works that apply PLM in CL point out the CF problem present in PLMs, there remain three significant issues that require further investigation. (1) Many existing CL works employ specific PLMs such as BERT (Devlin et al., 2019; Rogers et al., 2020), whereas more recent and sophisticated PLM structures, e.g. ALBERT (Lan et al., 2020) and XLNET (Yang et al., 2019), have been developed afterwards. A natural question would be whether the later PLMs could better mitigate CF in continual learning (Mosbach et al., 2021; Lee et al., 2020a). (2) The existing works focus on input, output, and gradients information of PLMs while ignoring the effect of model architecture on CF (Biesialska et al., 2020). We believe opening up the black-box of PLMs may lead to a deep understanding of their characteristics and thus better algorithmic design (Wallat et al., 2020). (3) Some existing works focus on alleviating forgetting in a trivial setting, during knowledge transfer between two tasks, which is not realistic in general continual learning with more tasks or even without task boundary (Jiang et al., 2020).

In this paper, we conduct an in-depth exploration of continual learning through extensive empirical analysis of a number of PLMs and CL methods. We analyze not only the performance differences across the combinations of PLMs and CL methods, but also the idiosyncrasies of each PLM.

Our main contributions are as follows:

- We design a rigorous benchmark for comprehensive study of continual learning in NLP. We conduct experiments over (1) two primary continual learning setting, including task-incremental learning and class-incremental learning; (2) three benchmark datasets with different data distributions and task definitions, including relation extraction, event classification, and intent detection; (3) four CL approaches with six baseline methods implemented for systematic comparison; and (4) five pretrained language models.

- We evaluate and contrast the performance of the above combinations of settings, PLMs and CL methods, providing a comprehensive comparative study from a number of perspectives.

- We dissect the performance characteristics of different PLMs with a number of layer-wise probing analyses.

- Our observations and insights give rise to a number of open research questions that can guide the design and optimization of better PLM-oriented continual learning methods and CL-oriented pretraining strategies.

- To encourage more research on continual learning in NLP, we release the code and dataset as an open-access resource on `https://github.com/wutong8023/PLM4CL.git`.

## 2 BACKGROUND

In this section, we provide the required background on pre-trained language models as well as continual-learning settings, methods and evaluation metrics.

### 2.1 PRETRAINED LANGUAGE MODELS

It now become the best practice to incorporate PLMs into NLP systems for many problems such as question answering, machine reading comprehension, summarization, to name a few. Usually, making use of PLMs in such systems leads to significant performance gains in (weakly) supervised learning. In this work, we investigate the use of PLMs in continual learning. We now briefly introduce five typical PLMs that are evaluated in our work.

**BERT** (Devlin et al., 2019) is the most representative PLM, which uses bi-directional deep Transformers (Vaswani et al., 2017) as the backbone. BERT adopts the Masked Language Modeling (MLM) and the Next Sentence Prediction (NSP) as the self-supervision tasks for pre-training.

**ALBERT** (Lan et al., 2020) ALBERT is a lite version of BERT which sets the parameters of all Transformer blocks shared across all layers and factorizes the embedding matrices into two small size of matrices. Instead of NSP, ALBERT predicts the order of two consecutive textual segments. Although ALBERT utilizes significantly less memory, fine-tuning it on downstream tasks can achieve close performance to that of BERT.

**RoBERTa** (Liu et al., 2019) RoBERTa has almost the same architecture as BERT, while it differs in terms of three training procedures. RoBERTa removes the NSP loss, trains a model with bigger size and longer sequences, and creates dynamic MLM masks vs static ones used in BERT.

**GPT2** (Radford et al.) Unlike the aforementioned PLMs which are all masked language models, GPT2 is an autoregressive language model predicting one token at a time from left to right. GPT2 is often used for natural language generation, while the aforementioned PLMs are mostly used for natural language understanding.

**XLNET** (Yang et al., 2019) Unlike the masked language models, GPT2 can not utilize the context from the backward side. XLNET is an autoregressive language model as well, while it resolves this problem by adopting a new objective called Permutation Language Modeling, enabling the model to take advantage of both forward and backward contexts.

## 2.2 CONTINUAL LEARNING SETTINGS

Continual learning (CL) focuses on the development of learning algorithms able to accumulate knowledge on non-stationary data. CL approaches are benchmarked on their ability to learn a sequence of tasks without forgetting previously acquired knowledge. They are typically evaluated on incremental classification settings.

Some popular incremental learning scenarios in CL (van de Ven & Tolias, 2019; Zeno et al., 2018) include: class-incremental learning and task-incremental learning. In the training time of these incremental learning settings, a CL algorithm experiences each task sequentially (only once), and is informed about the distribution shift (aka the task boundary). They, however, differ in their assumptions about the evaluation. In task-incremental learning, the model relies on the task identity (or a task label) to make its prediction. Conversely, in class-incremental learning, the methods have to perform task inference, implicitly (Aljundi, 2019) or explicitly (Lee et al., 2020b). We will cover both settings in our study. A lesser known setting, namely Domain-incremental learning, also relaxes the task-ID dependence, but shares the same output head for each task. For more details on these settings, see (van de Ven & Tolias, 2019).

## 2.3 CONTINUAL LEARNING APPROACHES

Many continual-learning approaches have been proposed recently, e.g., see Delange et al. (2021) for an overview, which can be categorised as follows:

**Rehearsal-based Approach**   The simplest way to reduce forgetting is to store samples from the past and reuse them to complement the learning of new tasks. In its simplest form, old samples are replayed with new data. This strategy, known as experience replay (ER), (Rolnick et al., 2019) is often a hard to beat baseline. Numerous rehearsal-based methods have been developed to increase ER's performance or efficiency (Aljundi et al., 2019a; Caccia et al., 2019; Hayes et al., 2018). Instead of replays, some methods use the old sample to perform constrained optimization to prevent increasing the loss on old tasks (Lopez-Paz & Ranzato, 2017; Aljundi et al., 2019b).

**Regularization-based Approach**   Also know as *prior-based* approaches, these methods prevent significant updates to the parameters that are deemed important for the previous tasks. Their first instantiation appeared in the Elastic-Weight Consolidation (EWC) (Kirkpatrick et al., 2017), where the previously learned weights are restrained from drifting via an L2 regularization loss. Regularization-based strategy often rely on the task boundaries to consolidate their knowledge during training. They often fail on long tasks sequences or settings where the task identity is not observable (Farquhar & Gal, 2018; Lesort et al., 2019b; Chaudhry et al., 2018). Despite these findings, prior-focused methods are actively researched (Zeno et al., 2018; Nguyen et al., 2018).

**Dynamic Architecture Approach**   Also know as *parameter-isolation* methods, this family of algorithm alleviates forgetting by using different subset of parameters for fitting different tasks. One popular approach, called Hard Attention to the Task (HAT) (Serrà et al., 2018), achieves parameter freezing through an attention mask that is learned concurrently at every tasks. Other similar strategies have also been proposed (Yoon et al., 2017; Schwarz et al., 2018). Similarly to regularization-based methods, dynamic architectures usually assume the availability of test-time task labels, and thus are not straightforwardly applicable in more realistic settings. A notable exception can be found in Ostapenko et al. (2019).

**Hybrid Approach**   Some hybrid strategy from different strategy also exist. E.g. Dark Experience Replay (DERPP) Buzzega et al. (2020) proposes a method at the intersection of rehearsals and regularization based methods.

The rapid growth of continual learning has lead researchers to work on empirical studies (De Lange et al., 2019; Lesort et al., 2021b), surveys (Hadsell et al., 2020; Khetarpal et al., 2020; Lesort et al., 2021a; Mundt et al., 2020; 2021) as well as CL-specific software (Normandin et al., 2021; Douillard & Lesort, 2021; Lomonaco et al., 2021). Further approaches are detailed in Appendix A.

## 3 BENCHMARKING CONTINUAL LEARNING OF PLMS

In this section, we first compare the performance of different combinations of CL methods and PLMs on three benchmark datasets, and we explore the following three research questions: (1) Does the catastrophic forgetting problem exist in PLMs during continual learning? (2) Which continual learning approach is the most efficient for PLMs and why? (3) Which PLM is the most robust for continual learning and why?

### 3.1 EXPERIMENTAL SETUP

**Methods**. We adopt 5 representative PLMs for evaluation, i.e., ALBERT (Lan et al., 2020), BERT (Devlin et al., 2019), GPT2 (Radford et al., 2019), RoBERTa (Liu et al., 2019), and XL-NET (Yang et al., 2019). By combining a PLM and a linear classifier as the backbone model, we investigate the following 4 approaches for the comparative study: (1) *Vanilla* uses the model learned on previous tasks as initialization and then optimizes the parameters for the current task. This baseline greedily trains the model on each task without accessing data from previous tasks, and is thus severely prone to catastrophic forgetting. It serves as a weak lower bound in terms of the average accuracy. (2) *Joint* trains on the entire training set simultaneously in the conventional supervised learning setup. Therefore, Joint does not suffer from forgetting and represents the performance upper bound. (3) *EWC* (Kirkpatrick et al., 2017) is a regularization-based methods, extending the loss function with a regularization term that slows down the updates of the important network weights. (4) *HAT* (Serrà et al., 2018) is a dynamic architecture method, employing a heuristic strategy to prevent intransigence by allocating additional units to the network when needed. (5) *ER* is a rehearsal-based method, interleaving old samples with current data in training batches. (6) *DERPP* (Buzzega et al., 2020) is a hybrid method, combining the strategy of rehearsal and regularization which prevents the prediction logits of memorized samples from changing. Following DERPP (Buzzega et al., 2020), we simulate the sequence of tasks according to the class-incremental learning setting (i.e. Class-IL or CIL) and task-incremental learning setting (i.e. Task-IL or TIL), splitting the training dataset into partitions of classes/tasks. Training details and hyper-pearameters are in Appendix C

**Metrics**. To measure the FWT and BWT abilities of the CL models, we assume access to a test set for each task. After the model learning on the training set of task $t$, we present the evaluation results on all $T$ tasks. We adopt three evaluation metrics as in (Lopez-Paz & Ranzato, 2017): (1) Forward transfer $FWT = \frac{1}{T-1}\sum_{i=2}^{T-1} A_{T,i} - \tilde{b}_i$; (2) Backward Transfer $BWT = \frac{1}{T-1}\sum_{i=1}^{T-1} A_{T,i} - A_{i,i}$; And average accruacy $Avg.\ ACC = \frac{1}{T}\sum_{i=1}^{T} A_{T,i}$, where $A_{t,i}$ is the accuracy of models on the test set of $i$th task after model learning on the $t$th task and $\tilde{b}_i$ is the test accuracy for task $i$ at random initialization. In addition to FWT and BWT, we consider average accuracy as a key performance measure, measuring the accuracy of past tasks after the model has moved on to learning new tasks.

**Datasets**. We evaluate our methods on 3 datasets with distinct label distributions, covering the following domains. *CLINC150* (Larson et al., 2019) is an intent classification dataset with labels evenly distributed, including 150 classes and 100 instances per class. *Maven* (Wang et al., 2020) is a long-tailed event detection dataset with 163 classes, each of which is labelled at least 15 instances, reaching a total of 47,921 instances. *WebRED* (Ormandi et al., 2021) is a *severely* long-tailed relation classification dataset with 243 classes, each of which is labelled with at least 15 instances, reaching a total of 31,441 instances. Adhering to (Wu et al., 2021), we randomly split each dataset into disjoint tasks, and each tasks contains 10 classes. For each class, we randomly split the dataset set into train, validation and test set by 10:2:3. To reduce the amount of computation, we cap the number of instances per class for both Maven and WebRED to 1000. The data distributions of the three datasets is visualized in Appendix B.

### 3.2 RESULTS AND DISCUSSIONS

The main evaluation results over all studied PLMs and CL methods, in terms of accuracy, are summarized in Table 1. For the Task-IL setting, Figure 1 shows a comparison between the PLMs (a–d) and between CL methods (e–h) in terms of accuracy, backward transfer, forward transfer, and training time. Performance measure is averaged across the three datasets. From these results we can

| Settings | | Class-IL | | | | | Task-IL | | | | | |
|---|---|---|---|---|---|---|---|---|---|---|---|---|
| PLMs (Parameters) | | Vanilla | EWC | ER | DERPP | Joint | Vanilla | EWC | HAT | ER | DERPP | Joint |
| ALBERT(11.78M) | C | 6.04 | 6.29 | 63.07 | 73.24 | 94.58 | 15.24 | 14.09 | 19.96 | 77.96 | 86.96 | 96.84 |
| | M | 6.25 | 6.21 | 36.56 | 36.43 | 83.28 | 16.86 | 14.34 | 22.31 | 57.40 | 55.42 | 92.29 |
| | W | 3.41 | 3.80 | 22.65 | 19.52 | 61.19 | 14.66 | 11.94 | 16.95 | 47.21 | 43.7 | 93.33 |
| BERT(109.57M) | C | **15.09** | **12.11** | 82.13 | **86.29** | 95.69 | **51.22** | **41.71** | 34.56 | **92.00** | **94.71** | 97.07 |
| | M | **7.62** | 6.85 | **57.06** | 50.38 | **99.81** | **31.24** | 24.18 | 33.53 | **39.74** | **66.40** | **99.98** |
| | W | 4.21 | 4.02 | 36.11 | 29.42 | **69.78** | 15.75 | **21.00** | 21.11 | 59.86 | 47.47 | **95.56** |
| GPT2(124.53M) | C | 9.78 | 9.76 | 74.47 | 83.67 | 95.18 | 32.11 | 30.76 | 27.62 | 86.58 | 92.20 | 97.00 |
| | M | 6.27 | 6.26 | 50.17 | 46.31 | 93.39 | 20.99 | 23.54 | 20.25 | 69.70 | 62.82 | 99.24 |
| | W | 4.10 | 4.53 | 29.77 | 22.64 | 66.06 | 19.60 | 20.50 | 16.81 | 58.26 | 46.83 | 93.41 |
| RoBERTa(124.74M) | C | 14.89 | 11.20 | **83.04** | 83.62 | **96.31** | 50.76 | 37.18 | 38.09 | 91.84 | 90.38 | **97.89** |
| | M | 7.00 | 6.85 | 52.35 | 46.49 | 99.46 | 26.90 | 28.97 | 33.24 | 72.01 | 62.10 | 99.96 |
| | W | **5.15** | 3.99 | 36.24 | 35.10 | 71.89 | **23.88** | 17.08 | 19.31 | 61.99 | 54.41 | 95.06 |
| XLNET(116.81M) | C | 8.53 | 10.20 | 72.22 | 78.67 | 95.64 | 36.51 | 35.71 | **48.67** | 87.64 | 89.38 | 97.38 |
| | M | 6.40 | **8.98** | 53.26 | 46.81 | 99.69 | 24.76 | **32.15** | **39.74** | 75.12 | 61.19 | 99.96 |
| | W | 4.90 | **4.13** | **39.36** | **30.13** | 68.81 | 22.82 | 20.73 | **24.43** | **64.28** | 44.49 | 95.28 |

Table 1: Accuracy on benchmark datasets with two continual learning setting, where "C" is sequential CLINC150 dataset, "M" is the sequential MAVEN dataset, and "W" is the sequential WebRED dataset.

make the following observations on the three research questions that we posed previously. More analysis can be found in D and F.

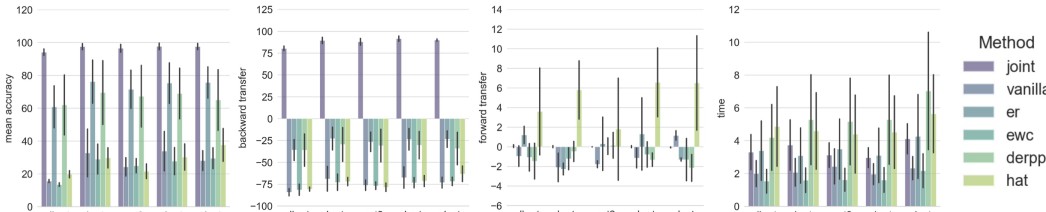

(a) Overall evaluation results in task-incremental learning grouped by PLM, including (1) accuracy↑, (2) backward transfer↑, (3) forward transfer↑, and (4) time↓.

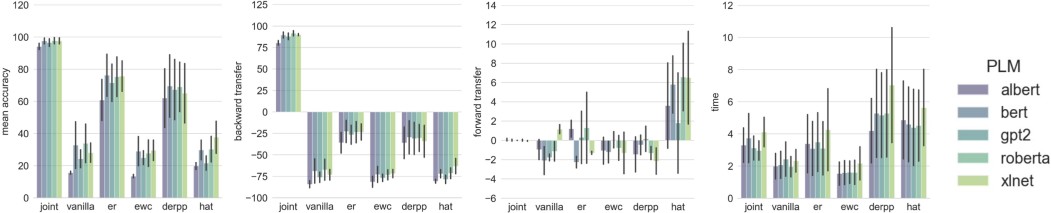

(b) Overall evaluation results in task-incremental learning grouped by continual learning method, including (1) accuracy↑, (2) backward transfer↑, (3) forward transfer↑, and (4) time↓.

Figure 1: Performance comparisons between PLMs (first row) and CL methods (second row) under the Task-incremental setting. For each row, the upper row is the results grouped by PLMs and the lower row is the results grouped by CL methods. For each column, from left to right the y label is accuracy, backward transfer, forward transfer, and time (in hours) respectively. Performance results are averaged across the three datasets. Note that the time spent by Joint in (a) and (b) is the time spent on training with the entire training set once.

For PLMs, catastrophic forgetting is serious. For each PLM, the performance gap between Vanilla and Joint indicates the PLM's tendency towards forgetting. For all PLMs on all three datasets, we can consistently observe significant performance gaps between the two methods, where the gaps are even more pronounced in Class-IL. This observation indicates that, without specialised algorithms, a direct adoption of PLMs in a CL environment will result in severe performance penalty.

**Imbalanced data is natural yet harder in continual learning than in conventional supervised learning.** Comparing the performance on different datasets, we notice that PLMs often achieve

better performance on more evenly distributed data (i.e. CLINC150 better than Maven better than WebRED). Compared to the conventional supervised setting (Joint), continual learning methods, including the well-performing methods such as ER and DERPP, are much more negatively impacted by data imbalance. This can be more easily observed in Figure 1(b-1).

Experience replay is the most robust method while regularization is not. Comparing the four different CL methods, as shown in Figure 1(a-2), we can observe that the methods adopting memory-replay (ER and DERPP) achieve much higher accuracy and higher backward transfer than the others (EWC and HAT). For dynamic architecture-based methods, HAT is the only method benefiting from continual learning, as it shows the positive Forward Transfer (FWT) in Figure 1(a-3, b-3). However, the higher training time, the lower accuracy, and the requirement for task label (thus inapplicable for Class-IL) do limit the applicability of HAT. Counterintuitively, the regularization-based method EWC shows the worst performance in our experimental setting. DERPP, which adopts both regularization and experience replay, reaches the highest accuracy on CLINC150. Surprisingly, it performs worse than ER on the other two imbalanced datasets (Figure 1(b-1,2)), which indicates that regularization-based methods may not be as robust as those based on experience replay.

BERT is still a good option for continual learning. Comparing different PLMs, there is no obvious difference on average accuracy and backward transfer, as shown in Figure 1(a-1,2). RoBERTa and GPT2 show the lowest FWT values in Figure 1(a-3), which means that their representative ability has been interfered the most during continual learning. For XLNET, Table 3 shows that it conducts the lowest computation per instance yet Figure 1(a-4) shows that it spends the longest time in training. In comparison, ALBERT has the least parameters and approx. equal time to obtain a competitive performance. As shown in Figure 1(a), BERT is still a good choice for continual learning scenarios as it achieves the highest accuracy while being competitive on the other three metrics.

### 3.3 NEW RESEARCH QUESTIONS

The above results show that among all PLMs, BERT is the most robust, and that among all CL methods, those based on experience replay are the most performant.These observations lead to the following important research questions.

(1) What happens inside the black box of BERT during continual learning?

(2) What is the performance difference across PLMs and across the layers inside each PLM?

(3) Why are replay-based methods more robust than regularization-based methods?

(4) In which layers does replay make the most contributions?

We further explore these research questions in Sections 4 and 5.

## 4 MINING THE SECRETS OF LAYERS IN BERT

In this section, we dive into BERT to provide a detailed analysis on the severity of forgetting on each of its layers, addressing RQ (1) of Section 3.3. Performing such an analysis further allows us to gain an understanding of the possible reasons of the robustness of experience replay-based methods, addressing RQ (3) of Section 3.3.

Generally speaking, the representative ability of a PLM increases with the increase of layers since such a bottom-up process can gradually gain more task-specific features. Hence, we put forward two possible reasons to catastrophic forgetting on BERT. (1) The continual learning problem with the PLM's ability to retrain the inner representations, which leads to catastrophic forgetting. (2) The continual learning of incremental tasks has minor influence on PLMs, but the classifier layer is the bottleneck. To investigate the above assumptions, we designed the following *probing* experiments.

### 4.1 PROBING SETTINGS

To investigate the representation ability per layer of a PLM, we follow Saunshi et al. (2019) and propose a prototype-based probing method (See Appendix E), which takes the layer-wise mean representation of instances belonging to the same class in the test set as the prototype representation of the class, and re-classify test instances by distance-based classification for the layer.

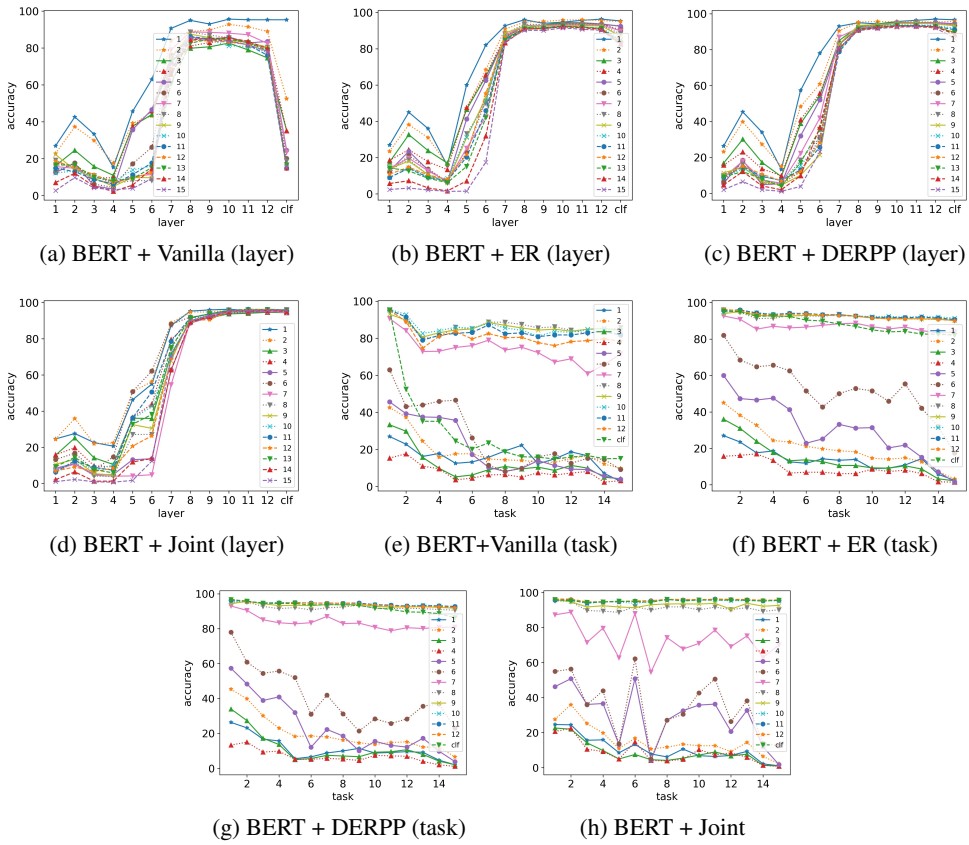

Figure 2: Layer-wise and task-wise probing of PLMs during continual learning, where $clf$ means classifier illustrating the prediction accuracy, number $k \in [1, 12]$ means representative of the k-th layer in a PLM.

Hence, we can use the mean classification accuracy as a short-hand for the representative ability for each layer, and obtain $T \times N$ results to track the performance per layer during continual learning, where $T$ is the number of learned tasks, i.e., time steps, and $N$ is the number of layers in a PLM, typically 12. To compare the the inner representative ability and the model output, we also track the prediction accuracy per task, which is denoted by $clf$. Note that, by comparing layer 12 and $clf$, we could learn the gap between the remaining representative ability of a PLM and the performance of a backbone model. To provide a fair comparison, we obtain the results for joint multi-task training by retraining a new backbone model from scratch at each time step, rather than retraining a model over seen data sequentially.

Figure 2 shows the probing results, in which we probe BERT with Vanilla, Joint and two CL methods based on experience replay, namely ER and DERPP. The analysis is done in two settings, layer-wise and task-wise.

**Layer-wise Probing.** In row 1, (a–d), the x-axis is the layers of BERT, and each line represents the accuracy after the model is trained with task $t$, as measured by the testsets of tasks 1 up to $t$.

**Task-wise Probing.** In row 2, (e–h), the x-axis represents the tasks, and each line represents the representativeness of each inner layer as well as the backbone model ($clf$), as measured by the testsets of tasks 1 up to $t$ after the model is trained with task $t$.

## 4.2 RESULTS AND DISCUSSIONS

Catastrophic forgetting occurs in both the last and middle layers. As shown in Figure 2(a–d), comparing the layer-wise performance of Vanilla to the other three methods, an obvious drop appears

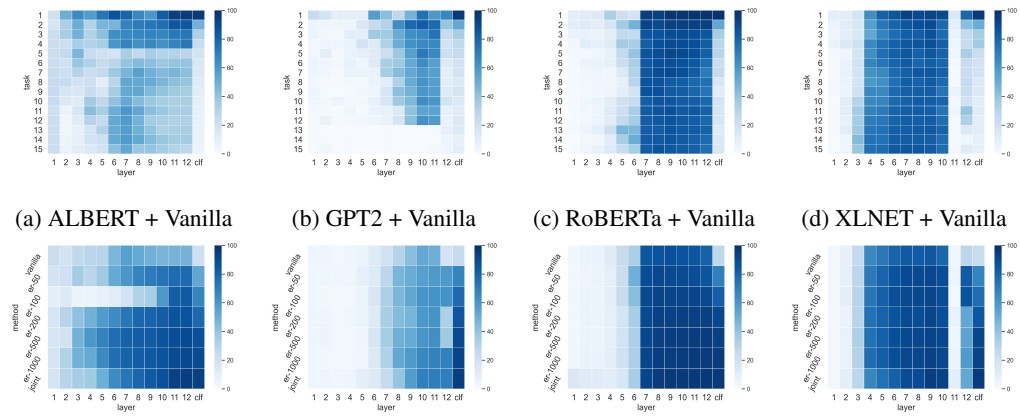

(a) ALBERT + Vanilla    (b) GPT2 + Vanilla    (c) RoBERTa + Vanilla    (d) XLNET + Vanilla

(e) ALBERT + buffer size    (f) GPT2 + buffer size    (g) RoBERTa + buffer size    (h) XLNET + buffer size

Figure 3: The buffer size analysis of ER. The first row is the detailed layer-wise analysis for different PLMs with Vanilla; The second row is the averaged layer-wise probing for ER with various buffer sizes. The color-scale represents the probing performance.

in Vanilla between layer 12 and the final classifier layer for all tasks except task 1. This result indicates that although BERT still maintains a high representative ability at the last time step, yet the classifier has already lost the ability to classify previously learned classes. Furthermore, as shown in Figure 2(e), the bottom (i.e. earlier) layers of BERT are consistently less representative than the top layers, and the drop is the most significant in the 6th and the classifier layers.

Regularization of outputs cannot prevent the forgetting in the bottom layers. Besides experience replay, DERPP also adopts regularization on the final classifier layer. Comparing ER and DERPP in Figure 2(f) and (g), DERPP can better constrain the $clf$ layer to maintain the representativeness of BERT, but it is less effective than ER on constraining the drop of the middle layers. Such differences may partially explain why ER could be more robust than regularization-based methods.

The layer-wise and task-wise probing experiments give rise to the following research questions that may shed lights on further improvements of CL methods. (1) The regularization of the middle layers may improve continual learning performance. (2) The magnitude of the difference between the last and penultimate layer shows that it is mostly a badly calibrated output layer that causes the forgetting.

## 5  UNDERSTANDING THE EFFECTIVENESS OF EXPERIENCE REPLAY

In this section, we probe the possible reasons behind the effectiveness of experience replay. We do so by trying to understand why BERT is robust among all PLMs (RQ (2) in Section 3.3), and by analyzing the effect of replay on different layers of PLMs (RQ (4) in Section 3.3). See Appendix G for more results.

### 5.1  PROBING SETTING

We extend the probing method in the above section to the following two probing settings.

**Layer-wise Probing for Vanilla.** As shown in Figure 3(a–d), we evaluate the layer-wise representativeness of different PLMs (except BERT) with Vanilla.

**Averaged Layer-wise Probing for ER with Various Buffer Size.** Figure 3(e–h) shows the mean accuracy per layer across time steps for ER with different buffer sizes, i.e., the number of seen training instances stored for the future replay. Doing so allows us to investigate on which layers do experience replay make the most contributions.

## 5.2 RESULTS AND DISCUSSION

The layer-wise representativeness varies significantly across PLMs. As shown in Figure 3(a-d), although the prediction of $clf$ is similar across PLMs, the performance characteristic of each PLM on their inner layers varies a lot. For example, due to its parameter-sharing mechanism, the hidden layers of ALBERT are more fragile than BERT (Figure 2(a)) and RoBERTa (Figure 3(c)), with larger accuracy gaps across tasks. Figure 3(e-h) shows that which layers benefit the most from the increase in buffer size differs across PLMs. ER improves the performance of layer 12 and $clf$ layers in XLNET, but for RoBERTa and GPT2 it is mainly the $clf$ layer.

Compared with the classification layer, the representation of some inner layers maintains a high performance. We notice a similar layer-wise performance curve in both BERT and RoBERTa, as shown in Figure 2(a) and Figure 3(c). In both cases, layers 7 to 12 seems to be robust during continual learning. For XLNET, the robust layers are 3 to 10, with a dramatic drop in layer 11. Moreover, some fragile layers can be observed for each PLM, such as layer 6 of BERT in Figure 2(e) and layer 12 for GPT2, and layer 11 for XLNET.

The robust layers in pretrained model may provide us the free lunch that we can extract features for metric-based classification without extra computation and buffer memory. The probing over five pretrained language model shows the existence of robust layer in BERT, XLNET, and RoBERTa. However, the post-selection of pretrained model and configuration may not always feasible in real-world applications, we believe it is an important direction to predict and dynamically detect the robust layer for better continual learning performance.

## 6 CONCLUSION

In this paper, we conduct the first comprehensive comparative study that sheds light on the performance characteristics of continual learning across representative language models and CL methods, as well as a detailed layer-wise analysis within language models. The insights gained from this study open up new research questions that will inspire further research on continual learning based on pre-trained self-supervised models.

Our comparative study on pretrained language models for continual learning provides more insights on different aspects of performance, uncovers new opportunities for the development of NLP-specific continual learning methods in which language models play a central role. Broadly speaking, the future impact of our work is threefold: (1) Criteria and strategies for selecting a PLM for continual learning. (2) Techniques for utilizing the layer-wise insights of a specific PLM for continual learning. Given the fragile layer and robust layer of a PLM, novel dynamic adapter structures, replay or regularization methods may be explored for more robust continual learning. (3) The open-source toolkit and best practices for consistent evaluation of CL performance.

## ACKNOWLEDGMENTS

Research in this paper was partially supported by the National Nature Science Foundation of China (U21A20488). The first author was partially supported by the Chinese Scholarship Council. The computational resources for this work were supported by the Multi-modal Australian ScienceS Imaging and Visualisation Environment (MASSIVE) (www.massive.org.au). We would like to thank the anonymous reviewers for their insightful comments.

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

## A    FURTHER RELEVANT APPROACHES TO CONTINUAL LEARNING

Instead of using previously-stored data for replay, some methods will train a generative model along-side their classifier and perform the rehearsal on generated samples. These methods are often re-ferred to as *generative replay* Shin et al. (2017); Lesort et al. (2019a); Ostapenko et al. (2019).

Furthermore, the field of meta-learning, now interrelating with continual learning, as provoked the emergence of new methods. In meta-continual learning Javed & White (2019); Vuorio et al. (2018), algorithms are learning how to continually learn. The hope is that the continual-learning problem can be solved in a data-driven way by the learning algorithm itself. In continual-meta learning, the task-agnostic CL setting, i.e., where the task boundaries aren't provided during training, is tackled. The algorithms perform fast adaptation Riemer et al. (2018) to adapt to the current task He et al. (2019); Caccia et al. (2020).

## B    DATASET DISTRIBUTIONS

The distribution of each dataset are summarized as Figure 4.

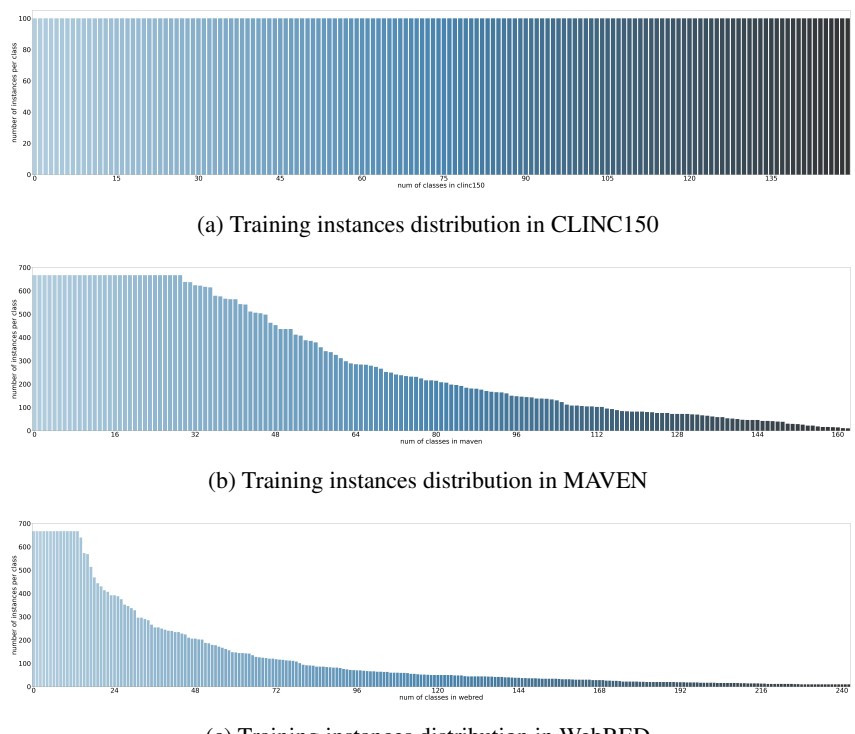

(a) Training instances distribution in CLINC150

(b) Training instances distribution in MAVEN

(c) Training instances distribution in WebRED

Figure 4: Data distribution

## C    MODEL TRAINING AND HYPER-PARAMETERS

**Hyperparameter selection**. For each combination of PLM and CL scheme, we perform a com-bined grid-search for Class-IL and Task-IL, choosing the configuration that achieves the highest final accuracy averaged on the two settings.

Typically, we summarize the details of implementation as follows:

**Training**. To provide a fair comparison among CL methods, we train all the networks using the AdamW Mosbach et al. (2021) optimizer, and select 10e-5 as the learning rate for all pretrained

| Method | Key | Value | Note |
|---|---|---|---|
| Vanilla | - | - | - |
| EWC | num_checkpoint | 1 | As the size of pretrained model is typically large, we only keep the check point from last task for regularization. |
| | $\lambda$ | 1,000,000 | The weight for panalty, selected from [0.1, 1, 10, 100, 1,000, 50,000, 1,000,000, 10,000,000] |
| | $\gamma$ | 0.2 | The weight for updating fisher information, selected from [0.1, 0.2, 0.3, 0.5, 0.7, 0.8, 0.9, 1]. |
| ER | buffer size | 200 | 500 for MAVEN and WEBRED. |
| | Sampling Method | Reservoir | Refer to DERPP |
| HAT | $\lambda$ | 0.7 | The weight for panalty, selected from [0.1, 0.5, 0.7, 1, 10, 100, ] |
| | $s_{max}$ | 400 | The scaling parameter for quantilization, refer to HAT. |
| | $cosh_{threshold}$ | 50 | The threshold for clamping, refer to HAT |
| | $embed_{threshold}$ | 6 | The threshold for embedding clamping, refer to HAT |
| DERPP | buffer size | 200 | 500 for MAVEN and WEBRED. |
| | Sampling Method | Reservoir | Refer to DERPP |
| | $\alpha$ | 0.5 | The weight for logits distillation, selected from [0.1, 0.5, 0.7, 1, 10, 100]. |
| | $\beta$ | 1 | The weight for experience replay, selected from [0.1, 0.5, 0.7, 1, 10, 100]. |

Table 2: The hyper-parameters of each baseline method.

| | ALBERT | BERT | GPT2 | RoBERTa | XLNET |
|---|---|---|---|---|---|
| GMac | 3.9 | 4.25 | —— | 4.25 | 2.83 |
| Parameters | 11.78M | 109.57M | 124.53M | 124.74M | 116.81M |

Table 3: Experimental computation complexity per instance.

backbone models. We deliberately hold batch size and minibatch size out from the hyperparameter space, thus avoiding the flaw caused by the different numbers of update steps for different methods.

# D    COMPUTATION COST

To provide a perspetive on computational complexity, we also measure the giga multiply-accumulate operations (GMac) per instance (spans of 50 tokens randomly selected from Maven) for each PLM and its number of parameters, as shown in Table 3.

# E    PROBING METHOD

As shown in Algorithm 1, we do layer-wise probing with the layer-wise prototype-based classification on validation dataset.

# F    FINAL STAGE PROBING

As shown in Figure 5 and Figure 7, pretrained language models show different layer-wise robustness at the last stage of task-incremental and class-incremental settings.

# G    OVERALL PERFORMANCE ON PLMS

As shown in Figure 7, pretrained language models demonstrate different task-wise representability during continual learning.

---

**Algorithm 1:** Function of Layer Evaluation $Evaluate\_Layer(\cdot)$

---

**Input:** Classification model $h_\theta$; Validation Dataset $\mathcal{D}^{val}$, Feature layers $\mathcal{A}_f$, Candidate Feature Layers $\mathcal{A}_c$.

1 Sample examples $(x_k, y_k) \in \mathcal{K} \subseteq \mathcal{D}^{val}$
2 Detach layer-wise hidden states $h_k^l \in \mathbb{R}^{|\mathcal{K}| \times |\mathcal{A}_c| \times d}$ for $x_k \in \mathcal{K}$ and $l \in \mathcal{A}_c$
3 Get layer-wise prototype $v_c^l$ via averaging $h_k^l$ by class
4 Get layer-wise distance $dist_k^l \leftarrow \sigma(h_k^l \cdot v_c^l)$ per example
5 Get layer-wise representability $acc_{\mathcal{A}_c} \leftarrow \frac{1}{|\mathcal{K}|} \sum_k \mathbb{I}(\arg\max[dist_k^l], y_k)$
6 **return** $acc_{\mathcal{A}_c}$

---

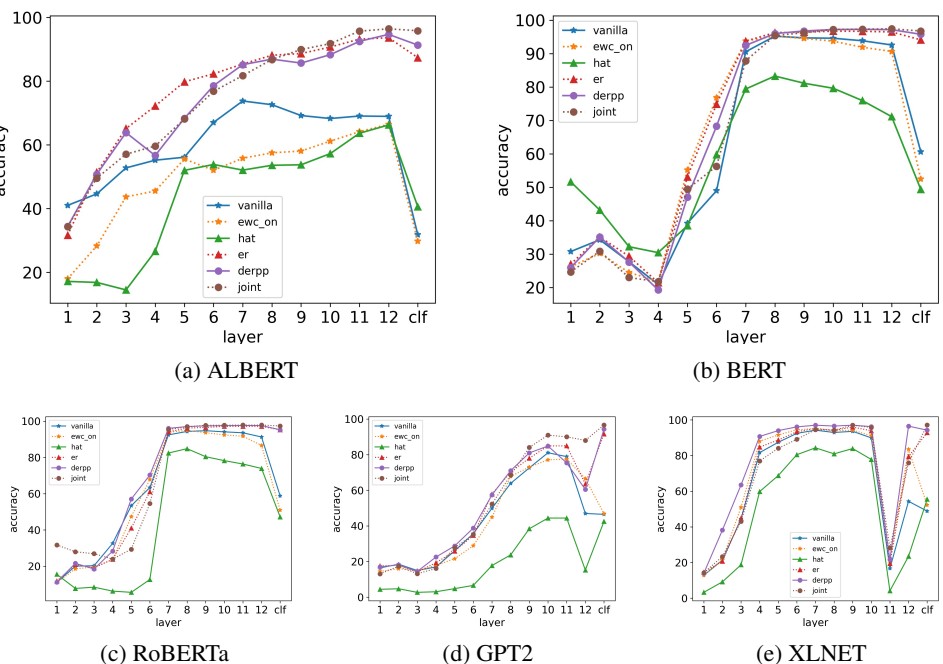

Figure 5: The layer-wise performance of PLMs during task-incremental learning.

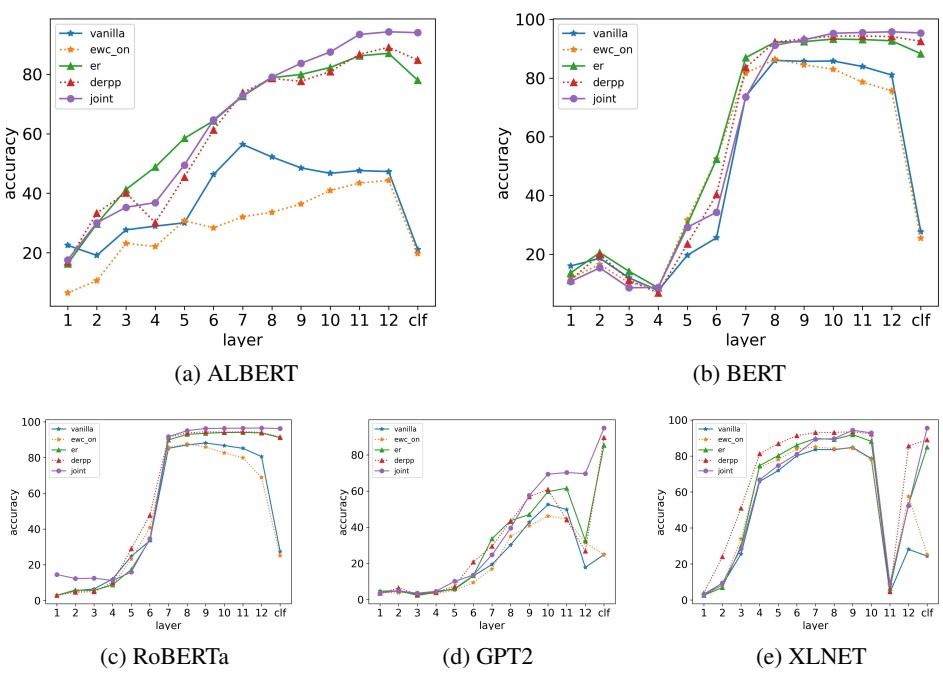

Figure 6: The layer-wise performance of PLMs at the last stage of class-incremental learning.

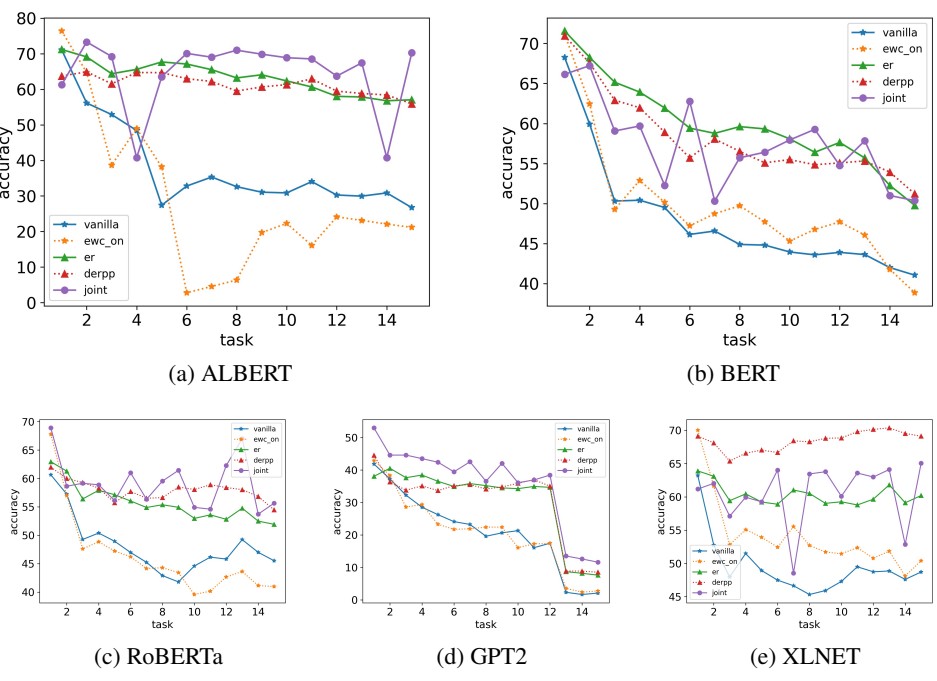

Figure 7: The task-wise performance of PLMs during class-incremental learning.

