# OpenReview forum: "Pretrained Language Model in Continual Learning: A Comparative Study"
_ICLR.cc/2022/Conference — ICLR 2022 Poster_

### Official Review · Reviewer_EuaL · 2021-10-31

**Correctness:** 4
**Technical Novelty And Significance:** 3
**Empirical Novelty And Significance:** 4
**Recommendation:** 8
**Confidence:** 5

**Main Review:**

## Strenghts:

The analysis of PLM is carried out over different continual learning techniques, NLP tasks, PLM variants and finally by layer-wise probing analysis. Understanding the strengths and weaknesses of each PLM is very desirable research progress. The authors also provide new research questions that arise from the analysis and point to interesting unsolved research challenges.

Evaluation starts off with the expected lower and upper bound of performance, and then moves on to disentangle FWT and BWT (backward, forward performance) when using various CL techniques.   The chosen data sets are not well behaved, i.e. experience imbalances, which makes the results more realistic (less artificial). Overall, this study provides a necessary step towards exploring future continual learning methodology and explores many important factors on eight pages.

From the batch-learning adaptation literature on PLMs one may expect baselines such as various adapter block versions or 'anti forgetting hacks', but it is understandable that the authors did not test these, since adapters would likely introduce complexity per increment and quickly become impractical. As the authors mention, CL specific future extensions to adapters are conceivable, but a work of their own.

### Minor weaknesses (easily fixable suggestions for improvement -- 1 content page left for improvements):

- Some plots seem to be a very small, and may be enlarged to use the 9th page.
- Fig 2 plots should share a larger model legend (on the figure top or bottom), so the bars can become wider and easier to distinguish
- Fig 2 color could be made more distinguishable, especially since the plots are narrow.
- Tab 1 could underline the best non joint performance — makes it easier to glance
- sec 4. That Transformer layers, except the classification layer, do not adapt much during fine-tuning, is a known result (hence assumptions 1 and 2 in the paper), which should be cited — see BERTOLOGY Primer by Anna Rogers for references. Here intermediate layers are shown to forget as well, so this is a nice new finding, that can be contrasted.
- Fig 3: I assume the figure color-scale is probe performance? Also, the buffer size, e.g. er-200 should be explained with an example. Is it 200 er samples, 200 samples/ class? In section 5.1 it should be (re-)mentioned what the buffer size means.

### Spelling errors (not all listed): using a TTS app/function makes it easy to find these

- sec 2.3. "eg".
- sec 3.2:
    - "could be find" ... can be found
    - tendency to forgetting ... to forget/ towards forgetting
    - reach the highest .. reaches
- 4
    - interferes ... with the PLMs ability to retrain _ representations
- 4.1
    - Joint ... Joint multi-task training
    - from sketch ... from scratch
- 4.2.
    - methods making sense ... make sense
    - The sentence "(2) The classification layer clf is typically the most fragile of BERT, where continual learning learning methods making sense." is not understandable.

## Questions to the authors:
None, regarding clarity.


**Summary Of The Paper:**

This paper explores the continual learning performance when combining different PLMs and common continual learning methods with 3 challenging NLP classification tasks.

To benchmark these combinations the methods are evaluated in task-incremental and class-incremental learning settings over various NLP end-tasks, which covers common learning settings in continual learning and NLP. There is also a layer-wise performance analysis to identify which layers keep or forget task relevant information during training.

Overall the paper shows that forms of replay outperform other methods like regularization.

**Summary Of The Review:**

The problem, proposed CL algorithms, benchmarks and evaluation metric are suitable to answer the research questions. The performance and layer-wise analysis is conductive to deepen an understanding for the shortcomings and potential opportunities of PLMs for continual learning. Unsurprisingly, variants of ER are the most effective technique in incremental CL with PLMs, which is somewhat disappointing, but also a standard outcome in CL.

The paper is mostly well written, and the experiments (sub research questions) are logically structured. Some minor details can be improved and have been pointed out in the review. Insights are valuable and provide a solid foundation for followup studies. The details in the appendix were interesting and added to the paper and its reading flow, e.g. by moving cumbersome details like hyperparameters to a dedicated appendix section.

I thus feel confident to recommend this paper for acceptance.

---

> ### Author Response · Authors · 2021-11-23
> **Response to Reviewer Eual**
>
> Dear reviewer,
> Thanks for the encouraging reviews and constructive criticism.
>
> The reviewer is right in pointing that the plots are too small for proper readability. We enlarged them, as well as font size and legends. And we will keeping work on Figure 2 to make it as clear as figure 3.
>
> We bolded the results to highlight the best performing methods. Thanks for the suggestion.
>
> We clarified the confusion related to Figure 3. Thanks for pointing it out.
>
> We bridged BERTOLOGY Primer to our work and mention it in the introduction.
>
> Please see our general reply for a new experiment  [\[**LINK**\]](https://openreview.net/forum?id=figzpGMrdD&noteId=f3FRIcTRJVF) that further illustrates our insight (See Appendix H). We will work it to the main paper in the next version.
>
> Thank you for pointing out the typos. We have fixed them in the current revision. Many thanks!
>
> Best regards,
>
> Authors

---

### Official Review · Reviewer_XW45 · 2021-11-03

**Correctness:** 3
**Technical Novelty And Significance:** 2
**Empirical Novelty And Significance:** 3
**Recommendation:** 6
**Confidence:** 4

**Main Review:**

Strengths:
Overall the study is very thorough covering both the correct range of options for each axis studied and a set of relevant cross-axis multiple variable experiments. The organization is good (but not perfect, see below), its strengths are that the different options considered along each axis are clearly laid out ahead of time, with the exception of the continual learning strategies. The layer-wise analysis in particular is interesting and tells a coherent story, despite the challenges of displayed complicated 3D data. Overall, it seems that recently many studies compare quantitatively against multiple PLMs, which ultimately appear similar due to only slightly different performance numbers. This study's most successful contribution in my opinion is an exploration of the qualitative differences among PLMs in the continual learning setting.

Weaknesses:
- It would be really great to see how the insights after analysis can be used to improve performance. It's probably not absolutely required given the focus on probeing, but it would go a long way towards validating the insights.
- Adjusting the numbers to achieve the 5/4/3/2 cuteness gets slightly in the way of understanding, unfortunately. The main reason for this is that the "veins of CL methods" has a different number over the course of the paper, which makes it hard to identify when a given list of N things is a list of the "veins of CL methods". Specifically, in the abstract and intro this number is 4, in section 2.3 this number is 3, in section 3.1 this number is 6, in Table 1 this number is 5 (two different sets of 5), in Figure 1 this number is 6, and in Figure 2 this number is 4. For a paper that has so much going on and so many different lists of different sizes, keeping these consistent would make it much easier for the reader to understand at any point what exactly this given list of N items is referring to. Along the same lines, it would help to be consistent with the language around each set of N things. For example, the "veins of CL methods" are called at least "veins", "schemes," and "approaches" at different points.
- Section 3.2, Table 1, and Figure 1 are relatively weak in my opinion. What am I supposed to conclude? I can look at the table and see the different results, but so what? What should I be drawing my eye to in the table (bold would help)? The Figure here is too small to even attempt to parse.
 - It would be very helpful to have a sentence that gives intuition about what's being measured with the accuracy metric. The definition is there, but it took me a second to realize that the intuitive idea is that it's measuring the accuracy of past tasks after the model has moved on to learning new tasks.


**Summary Of The Paper:**

The authors perform a comprehensive study of how pretrained language models work in the continual learning setting. The authors study 5 relevant pretrained language models (masked and unmasked) and somewhere between 3 and 6 continual learning strategies depending on where in the paper they are counted. In addition to a thorough everything-by-everything evaluation, the authors hone in on the details of how the different models and CL approaches are reflected in the transformer layers. The authors find that the different language models studied perform relatively differently, both qualitatively and quantitatively, and these insights may provide useful for directing future improvements.

**Summary Of The Review:**

Overall the authors perform a quite deep study of using pretrained language models for continual learning. Despite some weak points in the analysis of the quantitative results and inconsistent organization/language around the CL approaches, the thoroughness of the study, in particular the analysis at a layer-by-layer level, is likely of interest to the broader community.

---

> ### Author Response · Authors · 2021-11-23
> **Response to Reviewer XW45**
>
> Dear reviewer,
> Thanks for the encouraging reviews and constructive criticism.
>
> > Comment 1: It would be really great to see how the insights after analysis can be used to improve performance. It's probably not absolutely required given the focus on probing, but it would go a long way towards validating the insights.
>
> A: Thanks for the suggestion to improve performance in light of the probing results. We include a new empirical analysis of a simple strategy that selects layers to show the effectiveness of our insight. Please refer to our general reply for more details [\[**LINK**\]](https://openreview.net/forum?id=figzpGMrdD&noteId=f3FRIcTRJVF).
> We have included this part as Appendix H, and we will work it to the main paper in the next version.
>
> > Comment 2: For a paper that has so much going on and so many different lists of different sizes, keeping these consistent would make it much easier for the reader to understand at any point what exactly this given list of N items is referring to.
>
> A: We agree that the changing number of CL methods might get in the way of understanding the paper. However, because not all methods can be applied in all settings, we either have to choose between lowering the total number of results or having an unequal amount of methods in each section. Although we opted for the latter, we have now clarified the text. Thanks for the suggestion to increase consistency. Again, we have adjusted the text.
>
> > Comment 3: Section 3.2, Table 1, and Figure 1 are relatively weak in my opinion. What am I supposed to conclude? I can look at the table and see the different results, but so what? What should I be drawing my eye to in the table (bold would help)? The Figure here is too small to even attempt to parse.
>
> A: The reviewer is right that our figures are too small to parse. We were able to greatly increase visibility given the extra page we weren’t using yet. We have also added bolded results to increase readability. We agree with the reviewer that the current manuscript doesn’t reflect the main conclusion of Section 3.2. Specifically, we found that BERT is the superior continual-learning PLM given that it achieves a reasonable performance  despite its smaller parameter count. We have adjusted the text accordingly.
>
> > Comment 4: It would be very helpful to have a sentence that gives intuition about what's being measured with the accuracy metric. The definition is there, but it took me a second to realize that the intuitive idea is that it's measuring the accuracy of past tasks after the model has moved on to learning new tasks.
>
> A: We have adapted the text to better explain the accuracy metric. Thanks for the suggestion.

---

> > ### Author Response · Authors · 2021-11-23
> > **Results of New Experiment**
> >
> > A simple experiment that shows ``free lunch'', i.e. PLMs' strong capability of remembering, based on the layer-wise probing in PLMs. The numbers represent testing accuracy over seen classes after training on each task on the CLINC150 dataset. As can be seen, PLMs are able to accumulate knowledge in some specific layers which are revealed in our layer-wise probing experiments. For example, the 9th layer in RoBERTa (the **bold** row) is robust for continual learning even without any replay or regularization.
> >
> > | Model   | Method             | task1 | task2 | task3 | task4 | task5 | task6 | task7 | task8 | task9 | task10 | task11 | task12 | task13 | task14 | task15 |
> > |---------|--------------------|-------|-------|-------|-------|-------|-------|-------|-------|-------|--------|--------|--------|--------|--------|--------|
> > |   | Vanilla            | 93.67 | 47.50 | 32.56 | 22.92 | 20.53 | 15.67 | 13.57 | 12.04 | 11.19 | 9.83   | 9.00   | 7.94   | 7.33   | 6.64   | 6.73   |
> > | ALBERT        | Vanilla (layer-11) | 79.33 | 66.33 | 59.89 | 54.00 | 49.93 | 48.50 | 35.48 | 34.79 | 40.63 | 32.43  | 30.67  | 38.19  | 34.23  | 36.83  | 34.38  |
> > |         | ER                 | 94.33 | 92.67 | 87.22 | 87.33 | 86.07 | 81.61 | 80.62 | 75.38 | 76.85 | 74.87  | 72.06  | 68.06  | 65.38  | 63.86  | 63.07  |
> > |         |                    |       |       |       |       |       |       |       |       |       |        |        |        |        |        |        |
> > |     | Vanilla            | 95.00 | 49.50 | 35.44 | 33.50 | 28.60 | 17.56 | 24.62 | 19.29 | 17.96 | 18.93  | 14.55  | 13.86  | 14.10  | 10.40  | 14.64  |
> > | Bert        | Vanilla ( layer-9) | 85.33 | 86.67 | 74.33 | 74.67 | 78.53 | 75.39 | 75.48 | 75.29 | 71.41 | 71.27  | 66.79  | 69.89  | 68.28  | 67.62  | 66.71  |
> > |         | ER                 | 95.33 | 95.00 | 92.44 | 92.75 | 92.40 | 90.39 | 89.90 | 88.25 | 86.96 | 85.03  | 83.94  | 84.22  | 82.72  | 82.71  | 82.13  |
> > |         |                    |       |       |       |       |       |       |       |       |       |        |        |        |        |        |        |
> > |    | Vanilla            | 95.00 | 55.67 | 34.22 | 34.08 | 28.33 | 19.11 | 17.57 | 14.71 | 14.44 | 16.07  | 16.67  | 13.50  | 14.97  | 8.83   | 10.58  |
> > | GPT2         | Vanilla (layer-10)  | 90.33 | 65.00 | 42.56 | 62.33 | 42.87 | 45.61 | 49.33 | 42.58 | 45.67 | 49.60  | 48.00  | 44.44  | 38.03  | 44.05  | 35.84  |
> > |         | ER                 | 94.67 | 92.00 | 89.00 | 91.33 | 89.47 | 88.61 | 86.38 | 86.50 | 84.89 | 80.97  | 81.45  | 81.36  | 81.51  | 77.24  | 74.47  |
> > |         |                    |       |       |       |       |       |       |       |       |       |        |        |        |        |        |        |
> > |  | Vanilla            | 98.00 | 53.50 | 32.67 | 30.92 | 24.00 | 18.94 | 20.05 | 16.67 | 17.44 | 15.13  | 12.45  | 11.61  | 11.38  | 11.55  | 11.09  |
> > | RoBERTa        | Vanilla (layer-9)  | **97.00** | **85.83** | **77.33** | **74.33** | **71.60** | **72.72** | **73.86** | **68.67** | **72.37** | **70.83**  | **67.24**  | **69.08**  | **69.92**  | **70.88**  | **71.27**  |
> > |         | ER                 | 97.33 | 96.50 | 93.33 | 94.42 | 94.13 | 93.50 | 93.33 | 92.00 | 90.48 | 88.97  | 88.67  | 87.03  | 88.21  | 85.79  | 83.04  |
> > |         |                    |       |       |       |       |       |       |       |       |       |        |        |        |        |        |        |
> > |   | Vanilla            | 95.33 | 50.00 | 34.56 | 36.42 | 22.53 | 18.94 | 18.43 | 14.00 | 13.70 | 11.20  | 11.67  | 10.47  | 10.62  | 9.21   | 8.53   |
> > | XLNET         | Vanilla (layer-9)  | 67.33 | 75.83 | 63.89 | 70.58 | 68.13 | 66.61 | 62.86 | 61.92 | 61.78 | 59.40  | 58.12  | 59.67  | 60.31  | 61.05  | 61.76  |
> > |         | ER                 | 95.33 | 93.33 | 90.44 | 91.58 | 91.87 | 90.22 | 88.71 | 86.46 | 84.63 | 83.80  | 79.82  | 77.19  | 76.49  | 72.64  | 72.22  |

---

### Official Review · Reviewer_7fZV · 2021-11-03

**Correctness:** 3
**Technical Novelty And Significance:** 2
**Empirical Novelty And Significance:** 2
**Recommendation:** 5
**Confidence:** 4

**Main Review:**

Although the authors have conducted quite a lot of experiments, the phenomena shown in experiment results is hardly surprising to me. It is not surprising that the pre-trained language models would have forgetting issues when fine-tuned on downstream tasks. It is also not surprising that rehearsal-based methods perform the best for pre-trained models.

Moreover, the paper draws a conclusion that BERT is the most robust one and is a good option if a continual learning process is going to be conducted. Based on this, the authors provide a few analyses on BERT’s ‘secret’ for continual learning. However, compared with other pre-trained models, I don’t see that BERT is significantly better than others given the figures and tables. I feel from the figures and tables, BERT and other models look similar. The authors didn’t give a comprehensive explanation on how they read such information or a concrete quantitative comparison to support this claim.


**Summary Of The Paper:**


This paper conducts an empirical study on the catastrophic forgetting of pretrained language models. On two continual learning settings (class incremental and task incremental), the paper evaluates multiple pre-trained models on different data sets, to see how severe the catastrophic forgetting issue is for these pre-trained models. Then the paper also tests the effectiveness of multiple continual learning methods on such pre-trained models and draws some conclusions.


**Summary Of The Review:**

A thorough empirical analysis with unsurprising conclusions

---

> ### Author Response · Authors · 2021-11-23
> **Response to Reviewer 7fZV**
>
> Dear reviewer,
> Thanks for taking the time to carefully read our manuscript.
> > Comment 1: It is not surprising that the pre-trained language models would have forgetting issues when fine-tuned on downstream tasks.
>
> A: The reviewer is right that forgetting in PLMs may not be surprising, but we did not suggest that it is the forgetting surprising.
>
> We mentioned ‘surprising’ twice, once in abstract considering the performance differences across PLMs and across CL methods. And another point is in section 3.2, “Surprisingly, it performs worse than ER on the other two imbalanced datasets”, with respect to that a hybrid (ER + distillation) method is not as robust as just replaying.
>
> Besides, please refer to our general reply on insights gained [\[**LINK**\]](https://openreview.net/forum?id=figzpGMrdD&noteId=f3FRIcTRJVF), in particular the new experiment that (1) demonstrates PLMs' strong capability to remember and (2) establishes a new, strong performance lower bound. It may not be surprising that some layers suffer from forgetting, but this experiment shows us the existence of robust layers inside pretrained language models. Based on this new awareness of robust layers which are revealed in the layer-wise probing analysis, our trivially simple technique brings the Vanilla method a huge performance improvement from **11.09** to **71.27** (See the bold row with ``RoBERTa layer-9’’). And the same robustness can be observed across all 5 PLMs.
>
> > Comment 2: However, compared with other pre-trained models, I don’t see that BERT is significantly better than others given the figures and tables. I feel from the figures and tables, BERT and other models look similar. The authors didn’t give a comprehensive explanation on how they read such information or a concrete quantitative comparison to support this claim.
>
> A: The reviewer is right that the current Table 1 is not clear enough to conclude BERT is significantly better than others. Hence, following the reviewer’s suggestion, we bolded the highest performance over all of the 33 combinations of (dataset, continual learning method-setting) to compare pre-trained language models (See Table 1 in the revision). Of all 5 PLMs compared, BERT has the best performance most frequently, in 18 of 33 settings, far more than the second best PLM (XLNet, havin the best performance in 10 of 33 combinations).  In the current version, we also include the total number of parameters involved in computation, and BERT has the second least amount of parameters among the 5 PLMs. Therefore, BERT is suggested due to its best performance (highest accuracy and robustness) and best efficiency (fewest parameters except ALBERT).

---

### Official Review · Reviewer_P6Qd · 2021-11-05

**Correctness:** 4
**Technical Novelty And Significance:** 2
**Empirical Novelty And Significance:** 2
**Recommendation:** 3
**Confidence:** 4

**Main Review:**

I think a comparative study paper should suffice at least two conditions to be considered for a publication at a venue like ICLR. First, it should present a novel view on the problem, and second, it should draw a novel conclusion out of the experiments. Although the paper could be a good survey for readers who want to learn about continual learning, I think its viewpoint is not new and its conclusion is not surprising. While it is helpful to know that rehearsal works better than regularization in most datasets, this is not entirely a surprising result. I think it is a common belief that rehearsal-based is more robust against catastrophic forgetting, while regularization-method is more space-efficient in that it doesn't have to store examples. The fact that the last layer suffers from catastrophic forgetting is also not a surprising result, given that the lower layers are known to encode linguistic features and the upper layers encode task-specific features.

**Summary Of The Paper:**

As the title suggests, the paper is a comparison of recent continual learning methods that prevent catastrophic forgetting and their effectiveness in some text classification tasks using popular pretrained language models such as BERT, RoBERTa, etc. The paper divides continual learning methods into three categories: (1) rehearsal-based, (2) regularization-based, and (3) dynamic architecture. The experimental results show that rehearsal based methods are superior to the other two, and also that BERT is generally better than other candidates. The paper then proposes a new probing techniques to find out what makes rehearsal-based method better and what's happening inside BERT. The paper finds that the last layer has the biggest catastrophic forgetting and lower layer is less impacted.

**Summary Of The Review:**

While the paper can be helpful for readers who want to learn about continual learning in text classification using pretrained language models, the paper does not seem to bring sufficient a novel viewpoint or conclusion to be published at ICLR.

---

> ### Author Response · Authors · 2021-11-23
> **Response to Reviewer P6Qd**
>
> Dear reviewer,
> Thank you so much for your review and suggestions. Below are our responses to your comments.
>
> > Comment 1: I think its viewpoint is not new and its conclusion is not surprising. While it is helpful to know that rehearsal works better than regularization in most datasets, this is not entirely a surprising result.
>
> A: What we studied in this paper is not only revealing that rehearsal-based methods are more robust than others, but also analyzing the property of different PLMs via rehearsal-based methods.
>
> > Comment 2: The fact that the last layer suffers from catastrophic forgetting is also not a surprising result, given that the lower layers are known to encode linguistic features and the upper layers encode task-specific features.
>
> A: Based on Figure 2(a), the catastrophic forgetting problem in BERT happens in not only the last layer but also the intermediate layer. As an empirical study, our results are contrastive to BERTOLOGY (Anna Rogers et al, 2020). The magnitude of the difference between the last and penultimate layer is what's interesting. It tells us that CF is probably not as bad as it seems: it's mostly a symptom of a badly calibrated output layer. Although this might not surprise the reviewer, it is an important finding for the community. This insight can inspire (1) new methods that would focus more on alleviating the forgetting in the last layer instead of uniformly on all layers and (2) new methods that would take advantage of the intermediate robust layers.
>
> Hence, based on the layer-specific properties of PLMs, we conduct a new experiment to further validate our insights. Please refer to our general reply on insights gained  [\[**LINK**\]](https://openreview.net/forum?id=figzpGMrdD&noteId=f3FRIcTRJVF), in particular the new experiment that (1) demonstrates PLMs' strong capability to remember and (2) establishes a new, strong performance lower bound. It may not be surprising that some layers suffer from forgetting, but this experiment shows us the existence of robust layers inside pretrained language models. Based on this new awareness of robust layers, our trivially simple technique brings the Vanilla method a huge performance improvement from **11.09** to **71.27** (See the bold row with RoBERTa layer-9). And the same robustness can be observed across all 5 PLMs.
>
> Just as mentioned by reviewer EuaL, our paper is a necessary step in exploring future continual learning methodology, which may shed light on the further improvements of continual language learning methods.

---

### Author Response · Authors · 2021-11-23
**General Reply**

We thank all reviewers for the time in reviewing this paper and the suggestions and questions which are very helpful for us to improve this work. We would like to make a brief description of some new experiments here:

Based on the insights we derived in the paper, we designed the following experiments: We train a Vanilla model as described in the paper, that is, without any continual learning strategies such as replay or regularization. The only addition is that we keep only ONE instance for each seen class (in each task), which serves as the prototype during evaluation.

During inference, we select the best-performing intermediate layer empirically, based on Figure 3 of our layer-wise analysis, in which the darker the color the better the layer. We obtain the representations of the stored per-class instances from the selected layer, and use them as prototypes for distance-based classification. Note that the representations of testing instances are also detached from the selected layer.

The table below summarizes the results of this trivially simple technique, “Vanilla (layer-X)”. As can be seen, as training progresses, strikingly, this simple technique does not forget much, while the basic Vanilla model suffers significant forgetting. Moreover, the performance gap with ER remains stable and relatively small from around task 10 onwards. Finally, for RoBERTa, the performance of the 9th layer (bolded) shows that RoBERTa stably accumulates knowledge during continual learning, as its evaluation results seem to converge at 70.

So, with almost no extra computation for a continual training strategy, this technique seems to indicate that there is “free lunch” from pretrained language models. Note that, based on this awareness of robust layers, our trivially simple technique brings the Vanilla method a huge performance improvement from **11.09** to **71.27** (See the bold row with RoBERTa layer-9). And the same robustness can be observed across all 5 PLMs.

This new experiment further validates the insights in our paper, that pretrained language models are indeed capable of remembering layer-specifically, and the Vanilla model simply is not taking advantage of this layer-specific capability. This insight establishes a new empirical performance lower bound to continual language learning. Moreover, the same layer-wise robustness may also exist in a pretrained Transformer-based visual model, which opens doors to exciting new possibilities for a broader area.

We have included this part as Appendix H, and we will work it to the main paper in the next version.

---

> ### Author Response · Authors · 2021-11-23
> **Results**
>
> A simple experiment that shows ``free lunch'', i.e. PLMs' strong capability of remembering, based on the layer-wise probing in PLMs. The numbers represent testing accuracy over seen classes after training on each task on the CLINC150 dataset. The buffer memory size of ER is always 200. As can be seen, PLMs are able to accumulate knowledge in some specific layers revealed in our layer-wise probing experiments. For example, the 9th layer in RoBERTa (the **bold** row) is robust for continual learning even without any replay or regularization.
>
> | Model   | Method             | task1 | task2 | task3 | task4 | task5 | task6 | task7 | task8 | task9 | task10 | task11 | task12 | task13 | task14 | task15 |
> |---------|--------------------|-------|-------|-------|-------|-------|-------|-------|-------|-------|--------|--------|--------|--------|--------|--------|
> |   | Vanilla            | 93.67 | 47.50 | 32.56 | 22.92 | 20.53 | 15.67 | 13.57 | 12.04 | 11.19 | 9.83   | 9.00   | 7.94   | 7.33   | 6.64   | 6.73   |
> | ALBERT        | Vanilla (layer-11) | 79.33 | 66.33 | 59.89 | 54.00 | 49.93 | 48.50 | 35.48 | 34.79 | 40.63 | 32.43  | 30.67  | 38.19  | 34.23  | 36.83  | 34.38  |
> |         | ER                 | 94.33 | 92.67 | 87.22 | 87.33 | 86.07 | 81.61 | 80.62 | 75.38 | 76.85 | 74.87  | 72.06  | 68.06  | 65.38  | 63.86  | 63.07  |
> |         |                    |       |       |       |       |       |       |       |       |       |        |        |        |        |        |        |
> |     | Vanilla            | 95.00 | 49.50 | 35.44 | 33.50 | 28.60 | 17.56 | 24.62 | 19.29 | 17.96 | 18.93  | 14.55  | 13.86  | 14.10  | 10.40  | 14.64  |
> | Bert        | Vanilla ( layer-9) | 85.33 | 86.67 | 74.33 | 74.67 | 78.53 | 75.39 | 75.48 | 75.29 | 71.41 | 71.27  | 66.79  | 69.89  | 68.28  | 67.62  | 66.71  |
> |         | ER                 | 95.33 | 95.00 | 92.44 | 92.75 | 92.40 | 90.39 | 89.90 | 88.25 | 86.96 | 85.03  | 83.94  | 84.22  | 82.72  | 82.71  | 82.13  |
> |         |                    |       |       |       |       |       |       |       |       |       |        |        |        |        |        |        |
> |    | Vanilla            | 95.00 | 55.67 | 34.22 | 34.08 | 28.33 | 19.11 | 17.57 | 14.71 | 14.44 | 16.07  | 16.67  | 13.50  | 14.97  | 8.83   | 10.58  |
> | GPT2         | Vanilla (layer-10)  | 90.33 | 65.00 | 42.56 | 62.33 | 42.87 | 45.61 | 49.33 | 42.58 | 45.67 | 49.60  | 48.00  | 44.44  | 38.03  | 44.05  | 35.84  |
> |         | ER                 | 94.67 | 92.00 | 89.00 | 91.33 | 89.47 | 88.61 | 86.38 | 86.50 | 84.89 | 80.97  | 81.45  | 81.36  | 81.51  | 77.24  | 74.47  |
> |         |                    |       |       |       |       |       |       |       |       |       |        |        |        |        |        |        |
> |  | Vanilla            | 98.00 | 53.50 | 32.67 | 30.92 | 24.00 | 18.94 | 20.05 | 16.67 | 17.44 | 15.13  | 12.45  | 11.61  | 11.38  | 11.55  | 11.09  |
> | RoBERTa        | Vanilla (layer-9)  | **97.00** | **85.83** | **77.33** | **74.33** | **71.60** | **72.72** | **73.86** | **68.67** | **72.37** | **70.83**  | **67.24**  | **69.08**  | **69.92**  | **70.88**  | **71.27**  |
> |         | ER                 | 97.33 | 96.50 | 93.33 | 94.42 | 94.13 | 93.50 | 93.33 | 92.00 | 90.48 | 88.97  | 88.67  | 87.03  | 88.21  | 85.79  | 83.04  |
> |         |                    |       |       |       |       |       |       |       |       |       |        |        |        |        |        |        |
> |   | Vanilla            | 95.33 | 50.00 | 34.56 | 36.42 | 22.53 | 18.94 | 18.43 | 14.00 | 13.70 | 11.20  | 11.67  | 10.47  | 10.62  | 9.21   | 8.53   |
> | XLNET         | Vanilla (layer-9)  | 67.33 | 75.83 | 63.89 | 70.58 | 68.13 | 66.61 | 62.86 | 61.92 | 61.78 | 59.40  | 58.12  | 59.67  | 60.31  | 61.05  | 61.76  |
> |         | ER                 | 95.33 | 93.33 | 90.44 | 91.58 | 91.87 | 90.22 | 88.71 | 86.46 | 84.63 | 83.80  | 79.82  | 77.19  | 76.49  | 72.64  | 72.22  |

---

> > ### Comment · Reviewer_EuaL · 2021-11-29
> > **Interesting results, but beware of overclaiming -- more questions raised than answered**
> >
> > I still think that the base evaluation of CL with PLMs this paper establishes has merit for the community. While, one may argue that this work has few novel insights and too many known insights ([hindsight_bias](https://en.wikipedia.org/wiki/Hindsight_bias)), it was thus far unclear: (I) whether batch learning behavior reproduces in task-incremental learning, (II) whether and how these batch learning 'problems' can modified for continual setups via CL methods, and (III) what role different PLM architectures play. Thus, I still hold that someone should run this study, and the authors did. *The data setup (benchmark) and evaluation setup are interesting and the fact that the authors chose imbalanced, long-tailed dataset is appreciable. Imbalances are a meaningful fit with continual learning, and too many works are based on a 'far from naturally developing balanced settings'.* Since it has a topic of discussion in other venue, I will only score this paper on its perceived value, and leave the decision of 'venue bar passing' to the chairs. There are however two points (A,B) that should be addressed:
> >
> > **(A) New experiment may be wrong and lacks necessary ablations to claim insights, do not publish! (Does not affect my score)**
> >
> > However, this new "layer experiment" lacks rigor and ablations on its parameters -- understandably because of time constraints on experiment quality. Currently the experiment (reply) reads as lacking rigor (potentially wrong) because:
> > (0) The layers for this new experiment are identified after seeing $T$ test sets. This means the layer choice (hyperparameter) is based on test time information $\\{1..t\\}$. To study continual learning we can not assume access to test time information, i.e. same requirement as in batch learning. The layer choice has to be based on training time information and should be analyzed as tasks emerge.
> > (1) "we keep one sample": This is very unclear. How does this change affect (skew) the evaluation score? Does this mean there is 1 sample replay? If it is replay, then it would be instructive to know what effect ER buffer size has on evaluation. Maybe 1-10 samples suffice?
> > (2) "free lunch": Free lunch means that there is one model that is optimal for any downstream task -- e.g. time series prediction with *Bert likely suffers from inadequate biases. FL does not mean, "a bit good at many things." Please drop the term for the evaluation.
> > (3) "accumulate knowledge": there is no evidence to suggest that knowledge is accumulated (added over time) -- one would have to look at per class performance gains over time to claim accumulation. An established interpretation could be that there is reduced forgetting, and thus better continual learning.
> > (0) For continual learning
> >
> > These results should not be included in the current form as they may well hurt the insight robustness of the paper and venue, should they be published. Otherwise a through explanation with sensible ablations would be needed to make conclusive statements. Since the new experiments were time-constrained (produced under stress) and can simply be omitted, I will keep my score (8), as I do not think the new results affect the old ones.
> >
> > **(B) Fig2 axis are confusing, textual description is better.**
> >
> > The axis in Fig2 are in places not correct/ confusing, as they do not match the description. The y-axis, "task", in (a-e) is confusing. There is accuracy, tasks and layers, which suggests a 3D, rather than a 2D plot. Should the y-axis read "accuracy after 100 (T) tasks"?

---

> > > ### Author Response · Authors · 2021-12-04
> > > **The additional results for the new experiment. We apply the layer-specific experience obtained from CLINC150 to the other two datasets. (PART1/2 - WebRED)**
> > >
> > > | Webred  | Method       | task1 | task2 | task3 | task4 | task5 | task6 | task7 | task8 | task9 | task10 | task11 | task12 | task13 | task14 | task15 | task16 | task17 | task18 | task19 | task20 | task21 | task22 | task23 | task24 |
> > > |---------|--------------|-------|-------|-------|-------|-------|-------|-------|-------|-------|--------|--------|--------|--------|--------|--------|--------|--------|--------|--------|--------|--------|--------|--------|--------|
> > > |         | Vanilla      | 93.60 | 45.06 | 28.88 | 22.86 | 18.62 | 15.18 | 13.42 | 11.50 | 10.41 | 9.23   | 8.55   | 7.51   | 6.97   | 6.13   | 6.33   | 5.51   | 5.03   | 4.83   | 4.53   | 4.62   | 4.45   | 3.94   | 3.71   | 3.41   |
> > > | ALBERT  | V-L-11 | 68.86 | 38.88 | 25.03 | 14.81 | 16.50 | 11.11 | 16.71 | 11.18 | 9.34  | 10.64  | 9.34   | 9.55   | 9.20   | 13.02  | 8.34   | 8.14   | 8.46   | 7.70   | 7.93   | 7.88   | 8.16   | 8.33   | 7.87   | 8.87   |
> > > |         | ER           | 91.07 | 88.79 | 73.76 | 62.42 | 60.79 | 50.36 | 48.35 | 42.65 | 35.41 | 29.72  | 34.18  | 32.25  | 32.09  | 29.64  | 24.94  | 24.26  | 24.46  | 26.46  | 28.38  | 20.52  | 21.12  | 16.57  | 19.46  | 22.65  |
> > > |         |              |       |       |       |       |       |       |       |       |       |        |        |        |        |        |        |        |        |        |        |        |        |        |        |        |
> > > |         | Vanilla      | 98.19 | 49.36 | 31.56 | 23.15 | 19.05 | 16.39 | 12.81 | 12.44 | 10.90 | 8.65   | 9.13   | 9.43   | 7.45   | 7.13   | 6.73   | 6.72   | 7.43   | 6.94   | 4.99   | 6.06   | 4.98   | 4.96   | 4.26   | 4.21   |
> > > | Bert    | V-L-9  | 78.86 | 60.37 | 58.20 | 42.78 | 49.05 | 36.22 | 29.20 | 37.26 | 26.93 | 28.76  | 25.91  | 25.82  | 23.23  | 27.45  | 24.43  | 27.43  | 26.12  | 30.36  | 27.61  | 27.94  | 26.26  | 24.69  | 22.86  | 21.06  |
> > > |         | ER           | 98.71 | 95.72 | 89.44 | 86.59 | 77.66 | 70.55 | 64.95 | 61.54 | 63.05 | 61.47  | 59.24  | 50.25  | 52.85  | 49.81  | 45.19  | 46.79  | 40.99  | 35.65  | 36.36  | 38.80  | 37.98  | 37.91  | 33.63  | 36.11  |
> > > |         |              |       |       |       |       |       |       |       |       |       |        |        |        |        |        |        |        |        |        |        |        |        |        |        |        |
> > > |         | Vanilla      | 86.54 | 44.65 | 31.42 | 23.03 | 16.46 | 16.36 | 13.92 | 11.65 | 10.58 | 9.21   | 8.59   | 8.62   | 7.37   | 6.65   | 6.32   | 5.78   | 5.63   | 5.17   | 5.10   | 4.53   | 4.29   | 4.44   | 4.01   | 4.10   |
> > > | GPT2    | V-L-10 | 90.45 | 69.46 | 55.12 | 48.58 | 41.42 | 36.61 | 34.49 | 28.57 | 24.34 | 26.88  | 26.60  | 25.67  | 26.31  | 24.10  | 21.65  | 23.50  | 22.13  | 23.13  | 19.79  | 20.05  | 21.53  | 21.07  | 21.07  | 19.60  |
> > > |         | ER           | 92.67 | 91.29 | 82.51 | 74.42 | 61.22 | 65.79 | 63.25 | 56.30 | 53.05 | 47.22  | 48.91  | 46.07  | 38.53  | 38.48  | 36.86  | 31.81  | 36.32  | 32.66  | 34.72  | 35.63  | 32.01  | 31.17  | 27.10  | 29.77  |
> > > |         |         |       |       |       |       |       |       |       |       |       |        |        |        |        |        |        |        |        |        |        |        |        |        |        |        |
> > > |         | Vanilla      | 90.59 | 47.33 | 32.10 | 24.49 | 19.24 | 15.72 | 13.68 | 11.31 | 10.16 | 9.92   | 8.17   | 7.77   | 7.86   | 6.94   | 6.51   | 6.82   | 5.70   | 5.67   | 7.55   | 6.27   | 5.14   | 6.33   | 3.95   | 5.15   |
> > > | RoBERTa | V-L-9  | **97.50** |**72.44** | **42.32** | **39.32** | **30.04** | **29.14** | **37.24** | **32.93** | **31.43** | **23.54**  | **24.71**  | **29.74**  | **28.64**  | **33.34**  | **28.21**  | **22.63**  | **25.06**  | **24.44**  | **25.39**  | **22.54**  | **25.62**  | **24.35**  | **26.64**  | **24.60**  |
> > > |         | ER           | 97.63 | 90.23 | 77.53 | 78.69 | 80.67 | 72.31 | 72.33 | 70.74 | 66.05 | 60.32  | 55.74  | 56.31  | 54.92  | 52.87  | 51.27  | 41.71  | 46.76  | 44.92  | 42.44  | 42.24  | 38.04  | 40.26  | 39.71  | 36.24  |
> > > |         |         |       |       |       |       |       |       |       |       |       |        |        |        |        |        |        |        |        |        |        |        |        |        |        |        |
> > > |         | Vanilla      | 96.05 | 51.20 | 33.92 | 24.07 | 21.86 | 15.91 | 14.87 | 14.51 | 13.47 | 12.92  | 11.67  | 8.23   | 7.68   | 8.97   | 8.09   | 6.95   | 6.16   | 7.49   | 5.67   | 5.43   | 4.48   | 4.28   | 6.76   | 4.90   |
> > > | XLNET   | V-L-9  | 90.45 | 69.46 | 55.12 | 48.58 | 41.42 | 36.61 | 34.49 | 28.57 | 24.34 | 26.88  | 26.60  | 25.67  | 26.31  | 24.10  | 21.65  | 23.50  | 22.13  | 23.13  | 19.79  | 20.05  | 21.53  | 21.07  | 21.07  | 19.60  |
> > > |         | ER           | 93.44 | 88.25 | 85.85 | 84.94 | 71.27 | 71.09 | 68.72 | 62.57 | 58.61 | 57.00  | 56.29  | 56.60  | 54.73  | 47.52  | 50.36  | 47.12  | 46.55  | 45.93  | 44.06  | 40.31  | 39.95  | 35.29  | 38.50  | 39.36  |

---

> > > > ### Author Response · Authors · 2021-12-04
> > > > **The additional results for the new experiment. We apply the layer-specific experience obtained from CLINC150 to the other two datasets. (PART2/2 - MAVEN)**
> > > >
> > > > | MAVEN   | Method  | task1  | task2 | task3 | task4 | task5 | task6 | task7 | task8 | task9 | task10 | task11 | task12 | task13 | task14 | task15 | task16 |
> > > > |---------|---------|--------|-------|-------|-------|-------|-------|-------|-------|-------|--------|--------|--------|--------|--------|--------|--------|
> > > > |         | Vanilla | 100.00 | 50.00 | 33.33 | 25.00 | 20.00 | 16.67 | 14.29 | 12.50 | 11.11 | 10.00  | 9.09   | 8.33   | 7.69   | 7.14   | 6.67   | 6.25   |
> > > > | ALBERT  | V-L-11  | 99.69  | 60.25 | 41.32 | 29.90 | 25.62 | 18.62 | 13.61 | 15.52 | 15.96 | 9.61   | 8.96   | 11.72  | 10.24  | 12.25  | 11.52  | 9.14   |
> > > > |         | ER      | 99.71  | 90.44 | 81.09 | 78.57 | 65.85 | 60.75 | 59.93 | 57.80 | 53.57 | 49.49  | 47.41  | 42.89  | 44.27  | 42.77  | 35.77  | 36.56  |
> > > > |         |         |        |       |       |       |       |       |       |       |       |        |        |        |        |        |        |        |
> > > > |         | Vanilla | 99.96  | 50.00 | 33.33 | 25.06 | 20.20 | 19.51 | 18.68 | 12.75 | 12.19 | 10.71  | 9.15   | 9.51   | 8.03   | 7.53   | 8.39   | 7.62   |
> > > > | Bert    | V-L-9   | **99.97**  | **82.64** | **70.66** | **67.68** | **55.85** | **53.10** | **52.59** | **55.13**| **52.30** | **46.63**  | **49.24**  | **43.13**  | **43.65**  | **39.50**  | **42.90**  | **43.53**  |
> > > > |         | ER      | 99.90  | 94.82 | 90.03 | 82.62 | 82.10 | 78.12 | 76.75 | 71.03 | 71.82 | 66.73  | 66.55  | 69.18  | 57.73  | 56.79  | 55.95  | 57.06  |
> > > > |         |         |        |       |       |       |       |       |       |       |       |        |        |        |        |        |        |        |
> > > > |         | Vanilla | 99.30  | 53.80 | 39.95 | 25.09 | 19.98 | 16.58 | 14.27 | 12.69 | 11.97 | 10.18  | 9.09   | 8.33   | 7.84   | 8.45   | 6.98   | 6.27   |
> > > > | GPT2    | V-L-10  | 98.91  | 68.33 | 55.53 | 33.18 | 43.53 | 38.22 | 34.10 | 27.19 | 29.00 | 32.70  | 22.79  | 18.76  | 23.96  | 23.69  | 19.00  | 15.11  |
> > > > |         | ER      | 99.31  | 95.74 | 90.30 | 84.16 | 82.37 | 77.61 | 74.91 | 78.06 | 69.70 | 68.54  | 65.87  | 58.90  | 60.08  | 55.07  | 55.58  | 50.17  |
> > > > |         |         |        |       |       |       |       |       |       |       |       |        |        |        |        |        |        |        |
> > > > |         | Vanilla | 99.84  | 50.00 | 36.22 | 26.90 | 21.50 | 19.33 | 15.37 | 14.66 | 11.29 | 12.24  | 9.31   | 8.89   | 7.91   | 8.87   | 6.92   | 7.00   |
> > > > | RoBERTa | V-L-9   | **99.86**  | **78.45** | **63.03** | **58.24** | **55.33** | **48.94** | **40.34** | **46.25** | **41.84** | **43.96**  | **46.63**  | **43.55**  | **37.75**  | **43.12**  | **37.49**  | **33.87**  |
> > > > |         | ER      | 99.75  | 94.61 | 91.28 | 89.14 | 82.46 | 84.62 | 75.49 | 73.47 | 69.58 | 67.96  | 63.96  | 61.38  | 60.91  | 57.42  | 57.00  | 52.35  |
> > > > |         |         |        |       |       |       |       |       |       |       |       |        |        |        |        |        |        |        |
> > > > |         | Vanilla | 99.92  | 50.27 | 35.67 | 25.50 | 21.00 | 17.86 | 17.13 | 13.65 | 15.66 | 14.07  | 10.77  | 9.13   | 9.72   | 8.76   | 8.21   | 6.40   |
> > > > | XLNET   | V-L-9   | 99.90  | 82.04 | 65.46 | 47.31 | 49.95 | 52.42 | 45.35 | 42.56 | 39.30 | 41.20  | 36.22  | 35.60  | 34.17  | 36.70  | 32.96  | 28.48  |
> > > > |         | ER      | 99.84  | 91.82 | 90.10 | 88.26 | 81.65 | 78.22 | 73.67 | 76.66 | 72.50 | 66.46  | 64.76  | 63.43  | 61.61  | 56.19  | 54.46  | 53.26  |

---

### Decision · Program_Chairs · 2022-01-20

**Decision:**

Accept (Poster)

**Comment:**

This paper presents a comparison and analysis of continual learning methods for pretrained language models. The authors categorise continual learning methods into three categories, those that use cross task regularisation, those that employ some form of experience replay of previous training examples, and those that dynamically alter the network architecture for each task. Evaluation results from representative examples of these three paradigms are then presented and analysed. In general methods that incorporate experience reply appear to perform the best, while analysis of the predictive power of individual layers of the pretrained models suggests that some network layers are more robust to catastrophic forgetting than others, and that this also varies across architectures (BERT, ALBERT, etc.).

In general the reviewers agree that this is a well conducted study that provides an interesting contribution to an important area of research. They also generally agree that the many of the results are unsurprising given the properties of the algorithms explored and prior work in this area. The main point of difference then becomes how valuable it is to present a thorough study of existing algorithms that confirms our assumptions. I believe that the current work raises enough interesting questions to make it a useful contribution to researchers working in continual learning. In particular the results analysing the relative differences in catastrophic forgetting across different layers in models suggests interesting avenues for follow on work.